# Natural Antiseptics in Veterinary Practice: Evaluation of Efficacy and Safety

**DOI:** 10.3390/pathogens14040321

**Published:** 2025-03-27

**Authors:** Askar Nametov, Rashid Karmaliyev, Bakytkanym Kadraliyeva, Kenzhebek Murzabayev, Laura Dushayeva, Kanat Orynkhanov, Karagulov Adilbay, Marat Magzhan

**Affiliations:** 1Institute of Veterinary and Agrotechnology, Zhangir Khan West Kazakhstan Agrarian Technical University, Zhangir Khan St. 51, Uralsk 090009, Kazakhstan; bkadralieva@mail.ru (B.K.); murzabaev.k@mail.ru (K.M.); uralsk-laura@mail.ru (L.D.); adilbaj79@mail.ru (K.A.); magzhan.marat98@mail.ru (M.M.); 2Veterinary and Zooengineering Faculty, Kazakh National Agrarian Research University, Abay Ave. 8, Almaty 050010, Kazakhstan; k_orynkhanov@mail.ru

**Keywords:** anolyte, ozone, shungite, propolis, natural antiseptic, disinfection, microorganisms

## Abstract

Antiseptics are essential in infection control within veterinary medicine and are widely used for wound care, dermatological treatments, and disinfection. Growing interest in eco-friendly antiseptics has led to research on natural formulations. The aim of this study is to evaluate the efficacy and safety of natural antiseptic agents in combating pathogenic microorganisms and their use in disinfection. This article presents the results of efficacy testing, particularly focusing on Shozan (anolyte + ozone + shungite), which demonstrated strong bactericidal activity against pathogenic bacteria such as *Brucella melitensis*, effectively inhibiting their growth across various concentrations and exposure times. In contrast, Shuprozan (anolyte + ozone + shungite + propolis) and Prozan (anolyte + ozone + propolis) did not exhibit significant antibacterial effects. No antiviral activity was detected against the lumpy skin disease virus and foot-and-mouth disease virus, and no fungicidal properties were observed against *Trichophyton verrucosum* and *Histoplasma farciminosum*. This study’s results confirm the high efficacy and safety of natural antiseptics.

## 1. Introduction

The widespread use of antibiotics and antiseptics has led to a significant challenge: the development of microbial resistance to these agents. The number of resistant microbial strains continues to increase, weakening local immunity and contributing to the emergence of atypical diseases. This, in turn, prolongs treatment durations and increases the risk of complications [1,2,3].

Antiseptics play a key role in the prevention and treatment of infectious diseases in both human and veterinary medicine. Based on their chemical composition, they are classified as organic or inorganic. In veterinary practice, they are extensively used for wound care, treatment of dermatological conditions, and disinfection of instruments and facilities. In recent years, there has been a growing interest in environmentally safe natural antiseptics, such as plant extracts and essential oils, which are gaining increasing popularity [4,5,6].

Natural antiseptics are particularly attractive due to their biocompatibility, low toxicity, broad-spectrum activity, and ecological safety. However, their use requires caution, as some plant-derived components may be toxic to certain animal species or trigger allergic reactions [7].

Despite the availability of a wide range of chemical disinfectants, chlorine-based compounds remain widely used in veterinary practice due to their disinfectant, bleaching, and detergent properties. Chlorine exposure at 1 mg/dm^3^ effectively eliminates many bacteria and viruses. However, at higher concentrations, chlorine can cause irritation, burns, tissue necrosis, and inflammation, damaging the respiratory mucosa. Prolonged exposure may lead to deeper airway inflammation and have a negative impact on cardiovascular function [8].

In recent years, increasing attention has been directed toward natural antiseptics with antibacterial, antiviral, and fungicidal properties. In veterinary medicine, the use of antiseptic formulations such as Shuprozan, Prozan, and Shozan is emerging as a promising approach due to their safety, low toxicity, and minimal side effects [6].

These formulations demonstrate high efficacy in treating animal infections due to their strong antibacterial properties. They exhibit broad-spectrum antimicrobial activity and pose a low risk of microbial resistance, which is particularly crucial in modern veterinary practice, where minimizing chemical exposure in animals is a priority.

Shuprozan, Prozan, and Shozan are primarily used for external applications in veterinary medicine. They are effective in treating skin infections due to their antiseptic and bactericidal properties. Shuprozan, which contains shungite, propolis, ozone, and anolyte, effectively cleanses tissues and provides infection protection. Propolis enhances immune responses and promotes tissue regeneration, ozone destroys pathogenic microorganisms, and anolyte exhibits antimicrobial properties while aiding tissue repair.

Prozan and Shozan, which also contain propolis, ozone, and anolyte, possess strong antiseptic and anti-inflammatory properties, making them highly effective for infection control in animals. These formulations not only exert antimicrobial effects, but also promote tissue regeneration, prevent infections, and accelerate wound healing [9,10,11,12,13,14].

Thus, natural antiseptics represent a promising direction in veterinary medicine, contributing to animal health and reducing environmental risks.

*Objective*: This study aims to evaluate the efficacy and safety of natural antiseptic formulations in combating some pathogenic microorganisms and their application in disinfection.

## 2. Materials and Methods

### 2.1. Characterization and Preparation of the Investigated Preparations

This study focused on antiseptic formulations containing natural components: Shuprozan (shungite + propolis + ozone + anolyte), Prozan (propolis + ozone + anolyte), and Shozan (shungite + ozone + anolyte).

The Shozan complex (shungite + ozone + anolyte) was obtained by infusing shungite in ozonated anolyte. The neutral anolyte ANK produced in the STEL-10N-120-01 mod.120 IIP unit by the electrochemical treatment of a sodium chloride solution (0.9 g/L) in distilled water is a colorless transparent liquid with a chlorine odor containing highly active oxygen compounds of chlorine. The main active substances of this metastable solution are chloroxygen and hydroperoxide oxidants. The total content of dissolved substances in the ANK anolyte, at an oxidant concentration of 500 mg/L, does not exceed 5.0 g/L. The ANK anolyte was stored under laboratory conditions at room temperature in a tightly sealed glass bottle, protected from direct sunlight, and used without prior preparation or dilution.

The ozonated anolyte “Ozan” was produced using a DICHO ozone generator (model TQ-Z08) by placing the device’s nozzle into a 1 L container filled with ANK anolyte for 10 min, resulting in an ozonated anolyte with a saturating ozone concentration of 1.5 mg/L.

To prepare the “Shozan” solution, 0.1 kg of shungite was washed several times with tap water. The shungite was then placed in a sterile glass bottle and filled with 1 L of ozonated anolyte. The bottle was tightly sealed with a plastic cap and infused for at least three days at room temperature. The resulting Shozan solution was transferred to another container and stored under laboratory conditions in a refrigerator at +4 °C in a tightly sealed glass bottle. It was used without prior preparation or dilution.

The Prozan complex (propolis + ozone + anolyte) was obtained by mixing equal volumes of propolis extract and anolyte. In total, 50 g of propolis was frozen in a household freezer at −18 °C and then ground with a grater. The ground propolis was placed into a 500 mL container of distilled water and infused in a water bath for 1 h, resulting in a 10% propolis extract. The extract was cooled to room temperature, filtered through sterile gauze, and stored in a tightly sealed glass bottle in a refrigerator at +4 °C. It was used without prior preparation or dilution. To obtain Prozan, 50 mL of anolyte solution was mixed with 50 mL of 10% propolis extract, and the resulting solution was ozonated for 10 min. The Prozan preparation was stored in a tightly sealed glass bottle in a refrigerator at +4 °C and used without prior preparation or dilution.

The Shuprozan complex (shungite + propolis + ozone + anolyte) was prepared by infusing shungite in a Prozan solution at a 10:1 ratio for three days. The solution was stored in a tightly sealed glass bottle in a refrigerator at +4 °C and used without prior preparation or dilution.

To determine the antibacterial, antiviral, and fungicidal efficacy of the antiseptics, mainly strains of Brucella (causative agent of brucellosis in ruminants) and Pasteurella (causative agent of pasteurellosis in animals) were used, as well as nodular dermatitis and foot-and-mouth disease viruses, and fungal strains of lymphangitis in horses and trichophytosis in cattle. The names of the microorganism strains are presented in Table 1.

For this study, the following materials were used: 24-well culture plates, disposable pipettes (1, 5, 10 cm^3^), disposable tips for dispensers (100–200 µL, 1000 µL), Nutrient agar, Nutrient broth, Meat Peptone Agar (MPA), Sabouraud agar, 70% ethanol, 3% hydrogen peroxide, incubator, single-channel automatic pipette (100–200 µL), single-channel automatic pipette (1000 µL), pipette controller for serological pipettes, Petri dishes, and sterile cotton swabs.

### 2.2. Determination of Antibacterial Activity of the Preparations

Meat Peptone Agar (MPA) was prepared according to the manufacturer’s instructions and dispensed into 24-well culture plates at a volume of 2 cm^3^ per well. A bacterial suspension was prepared from *Brucella melitensis* and *Pasteurella multocida* cultures, which were spread as a “lawn” on the MPA surface. The concentration of bacterial cells in the suspension was determined using the McFarland turbidity standard, which was 1.7 × 10^9^ cells/mL [15].

The tested preparations (Shozan, Prozan, Shuprozan) were used in their native form in the following volumes: 1000, 900, 800, 700, 600, 500, 400, 300, 200, and 100 µL. The bacterial suspension was mixed with the tested preparations at the specified concentrations, and the mixtures, along with controls, were incubated at 37 °C for 15, 30, and 45 min. After the incubation period, the samples were inoculated into the wells containing Meat Peptone Agar (MPA). The antiseptic agents’ effect was halted upon sample transfer to agar, where bacterial growth or inhibition was further observed depending on the effectiveness of the preparation.

Bacterial colony growth or its absence was monitored daily for 24, 48, and 72 h. The growth of bacteria was assessed visually based on the morphological characteristics of the colonies on the Meat Peptone Agar. This allowed for the evaluation of how the exposure time of the preparations influenced bacterial growth and which concentrations were most effective in suppressing bacterial proliferation.

For comparative evaluation of the tested preparations’ effectiveness, 70% ethanol and 3% hydrogen peroxide were used as controls. The criteria for microbial growth assessment were expressed as “+” (growth) or “-” (no growth) (Table 2).

The studies were conducted in triplicate to ensure the reliability of the obtained results.

Repeated incubation of bacteria was not performed, as the aim of this study was to examine the bacteriostatic properties of the tested formulations. This research focused on the direct effect of the formulations on bacteria on agar under the specified incubation time conditions.

### 2.3. Determination of Antiviral Activity of the Preparations

The propagation and titration of lumpy skin disease virus and foot-and-mouth disease virus were carried out using a primary lamb testicular cell culture (LT) and a continuous baby hamster kidney cell culture (BHK-21) [16]. The virus titer in the culture medium was calculated using the Muench method. The effect of the antiseptic preparations on the virus strain was assessed based on the presence or absence of the virus’s cytopathic effect (CPE) on the monolayer cell culture. Working solutions of antiseptic formulations were prepared from the stock solution using a maintenance medium (MM):-100 µL Shozan + 900 µL MM-200 µL Shozan + 800 µL MM-300 µL Shozan + 700 µL MM-400 µL Shozan + 600 µL MM-500 µL Shozan + 500 µL MM-600 µL Shozan + 400 µL MM-700 µL Shozan + 300 µL MM-800 µL Shozan + 200 µL MM-900 µL Shozan + 100 µL MM

The working virus dose (100–150 TCID/mL) was calculated based on the biological activity of the strain. For the interaction between the preparation and the working virus dose, 1 mL of virus suspension (working dose) was added to each tube containing 1 mL of the corresponding preparation dilution. A separate tube containing 1 mL of maintenance medium without the preparation served as the “negative control”. Additional controls included hydrogen peroxide and ethanol. The mixture was incubated at 37 °C for 45 min.

After the incubation period, 200 µL of the mixture was inoculated into the wells of a culture plate containing a monolayer cell culture. The plates were incubated at 37 °C with 5% CO_2_ until the cytopathic effect of the working virus dose became evident. Controls (hydrogen peroxide and ethanol) were cultured under the same conditions as the tested samples.

To determine the presence of a viral cytopathic effect, microscopic examination of the monolayer culture was performed and compared with the reference well (negative control) containing only the working virus dose. The minimum concentration of the preparation that inhibited the cytopathic effect of the virus was identified as the lowest concentration that prevented the virus from affecting the monolayer cells.

The experiments were conducted in triplicate to ensure the reliability of the results.

### 2.4. Determination of Fungicidal Activity of the Preparations

Prepared according to the manufacturer’s instructions, Sabouraud agar was poured into 24-well plates, 2 cm^3^ per well.

From the fungal culture (*Histoplasma farciminosum*, *Trichophyton verrucosum*), inoculated as a “lawn” on the surface of Sabouraud agar, a suspension was prepared in physiological saline. The cell concentration of the fungal suspension was determined according to the McFarland turbidity standard, which was 1.7 × 10^9^ cells/mL.

The preparations (Shozan, Prozan, Shuprozan) were used in native form in the following volumes: 1000, 900, 800, 700, 600, 500, 400, 300, 200, 100 μL.

As controls, the following were used: 70% ethanol and 3% hydrogen peroxide.

The number of fungal cells of the studied fungal cultures in each tube was 8.5 × 10^8^ cells/mL, while the volumes of the preparations and controls were 500 μL, 450 μL, 400 μL, 350 μL, 300 μL, 250 μL, 200 μL, 150 μL, 100 μL, and 50 μL, respectively.

The contact time of the fungal culture suspension with the preparations in various concentrations (1000 μL to 100 μL) was 45 min, 30 min, and 15 min.

After the incubation of the fungal suspension and preparations for the specified time, the samples were inoculated into the wells containing Sabouraud agar. Centrifugation was not performed. The fungicidal activity assessment was conducted by measuring fungal growth and visually checking for the presence or absence of fungal colonies. The evaluation criteria for fungal growth were expressed as “+” or “-” (Table 3).

The experiments were conducted in three replicates to ensure the reliability and reproducibility of the obtained results.

### 2.5. Determination of Antimicrobial Efficacy of Preparations in Industrial Conditions

The antimicrobial efficacy of the preparations Shozan, Prozan, and Shuprozan was evaluated under industrial conditions at the meat processing plant LLP “Zhaiyk Et”. To evaluate the antiseptic properties of Shozan, the solution was applied to various surfaces by spraying with a mechanical sprayer at a flow rate of 1 L per 5 m^2^, followed by a 30 min exposure. To assess the level of air contamination in the plant’s workshops, a preliminary microbial landscape was established using the sedimentation method. For this purpose, Petri dishes with Meat Peptone Agar were placed at sampling points on horizontal surfaces and left open for 5 min. The calculation was performed at a rate of one air sample per 20 m^2^ of surface area, following an envelope-type pattern: four points at the corners of the room at a distance of 0.5 m from the walls and a fifth point in the center. Samples were collected during the daytime, after wet cleaning and room ventilation.

To evaluate the microbial contamination of surfaces in production areas, swab sampling was conducted before and after treatment in six workshops of the LLP “Zhaiyk Et” plant. A total of 260 samples were analyzed, including 130 air microbiota samples from all workshops and 130 swab samples from various surfaces.

The studied workshops included the following:1.Live poultry handling2.Stunning and bleeding3.Scalding4.Sanitary processing5.Evisceration and butchering6.Packaging

Sterile test tubes containing 9 mL of sterile physiological saline were used for swab sampling. Before the start of hydro-cleaning, five swab samples were collected from each workshop after mechanical cleaning. These samples were pooled into a single composite sample per workshop. Samples were collected from surfaces including floors, walls, poultry hangers, and tables. Swabs were taken by thoroughly wiping a 10 × 10 cm area with a moistened cotton-gauze swab before and 5 min after antiseptic treatment. The swabs were then washed in 10 mL of sterile physiological saline, and 1 mL of the resulting suspension was transferred using a sterile pipette into a test tube containing 9 mL of sterile physiological saline.

To determine the total bacterial contamination, 1 mL of each swab suspension was inoculated into sterile Petri dishes, carefully lifting the lid slightly. After inoculation, 10 mL of molten Meat Peptone Agar, cooled to 45 °C, was poured into each plate after flaming the rim of the test tube containing the medium. The plates were immediately mixed to evenly distribute the sample across the entire dish surface. After solidification, the inoculated plates were incubated in an inverted position at 37 ± 1 °C for 24 h.

On the following day, all bacterial colonies that had grown in the Petri dishes were counted. The experiments were conducted in three replicates to ensure the reliability and reproducibility of the results.

## 3. Results

### 3.1. Antibacterial Activity of Shozan, Shuprozan, and Prozan Against B. melitensis and Pasteurella multocida

#### 3.1.1. Effect of Shozan on *B. melitensis*

The growth of *B. melitensis* (control) on Meat Peptone Agar was observed after 24 h of cultivation. Growth suppression of *B. melitensis* was recorded after 45 and 30 min of contact with Shozan at concentrations ranging from 1000 µL to 200 µL; 15 min of contact with the bacterial suspension resulted in growth inhibition at concentrations from 1000 µL to 400 µL.

After 48 h of cultivation, Shozan at concentrations from 1000 µL to 300 µL (with a 45 min of contact) inhibited the reproduction of *B. melitensis*; 30 min of contact suppressed bacterial growth at concentrations from 1000 µL to 400 µL; and 15 min of contact ensured inhibition at concentrations of 1000 and 900 µL.

After 72 h of cultivation, 45 min of contact with the preparation ensured the growth suppression of the bacterial culture at concentrations of 1000 and 900 µL.

#### 3.1.2. Effect of Shozan on *Pasteurella multocida*

When assessing the bacteriostatic activity of Shozan against *P. multocida*, it was found that Shozan at concentrations ranging from 1000 µL to 100 µL, with a microbial cell count of 1.7 × 10^9^/mL and contact times of 45, 30, and 15 min, did not exert an inhibitory effect on the growth of *P. multocida*. The results obtained using Shozan, compared to traditional antiseptics, are presented in Table 4.

In a comparative study with traditional antiseptics, it was found that 70% ethanol and 3% hydrogen peroxide effectively inhibit the growth of *B. melitensis* after 24 h of bacterial suspension cultivation. Ethanol at concentrations ranging from 1000 to 100 µL, with contact times of 15, 30, and 45 min, completely inhibited the growth of these microorganisms.

After 48 and 72 h of cultivation, it was established that 70% ethanol and 3% hydrogen peroxide at concentrations from 1000 to 500 µL, with contact times of 45, 30, and 15 min, suppressed the reproduction of *B. melitensis*. These results confirm that both substances exhibit antibacterial activity, effectively inhibiting the multiplication of these microorganisms under the specified conditions.

The antibacterial activity of 70% ethanol and 3% hydrogen peroxide against *P. multocida* was observed for 24 h at reagent volumes ranging from 500 to 1000 µL. After 48 and 72 h of *P. multocida* cultivation, the activity of 70% ethanol significantly decreased, remaining effective only at a volume of 1000 µL, whereas 3% hydrogen peroxide remained effective at all exposure stages.

When evaluating the antibacterial activity of the drugs Shuprozan and Prozan at concentrations ranging from 1000 to 100 µL at various time intervals (45, 30, and 15 min) against *B. melitensis* and *P. multocida*, at an initial microbial cell concentration of 1.7 × 10^9^/mL with 24 h cultivation, it was found that, under all specified conditions, the drugs did not affect the growth of these bacteria. This suggests that both drugs likely do not exhibit significant antibacterial activity against *B. melitensis* and *P. multocida* at these concentrations and exposure conditions. All findings were confirmed by results from three repeated experiments.

### 3.2. Determination of the Antiviral Activity of Shozan, Prozan, and Shuprozan Against Lumpy Skin Disease and Foot-and-Mouth Disease Viruses

In the study of the antiviral activity of Shozan, Prozan, and Shuprozan against the lumpy skin disease virus, it was found that these antiseptics do not exhibit significant activity against this virus (Table 5).

During the study of the antiviral activity of Shozan, Prozan, and Shuprozan against the foot-and-mouth disease virus, it was found that these antiseptics do not exhibit significant activity against this virus (Table 6).

The determination of the fungicidal activity of Shozan, Prozan, and Shuprozan against fungi of the genera *Trichophyton verrucosum* and *Histoplasma farciminosum* showed that the antiseptic formulations did not exhibit fungicidal activity against either strain. Both cultures continued to grow, and no signs of growth inhibition were observed (Table 7).

All tested preparations, including Shozan, Prozan, and Shuprozan, as well as traditional antiseptics—70% ethanol and 3% hydrogen peroxide—did not demonstrate fungicidal activity against *T. verrucosum* and *H. farciminosum* strains. This indicates their inability to effectively suppress growth or eliminate these fungal cultures at the tested concentrations.

### 3.3. Determination of the Antimicrobial Effectiveness of the Preparations in Industrial Conditions

When evaluating the antimicrobial effectiveness of Shozan, Prozan, and Shuprozan at the “Zhailyk Et” meat processing plant, it was found that the microbial composition of the air in the production workshops included *E. coli*, *Bacillus*, and *Staphylococcus aureus*. These microorganisms were also detected in swabs taken from various surfaces. The baseline total microbial count (TMC) on equipment in the workshops ranged from 9.98 × 10^3^ to 2.8 × 10^6^ CFU/100 cm^2^.

During industrial trials at “Zhailyk Et”, it was established that Shozan, Prozan, and Shuprozan effectively suppressed the growth of coliform bacteria and staphylococci after a 30 min exposure on all treated surfaces and in the air. After disinfecting the workshop surfaces with Shozan, microbial contamination (TMC) decreased to 3.35 × 10^2^–2.12 × 10^4^ CFU/100 cm^2^, while the use of Prozan reduced contamination to 9.2 × 10^2^–6.6 × 10^4^ CFU/100 cm^2^. The application of Shuprozan led to a reduction in contamination (TMC) to 9.4 × 10^3^–8.4 × 10^5^ CFU/100 cm^2^ (Table 8).

Thus, the highest antimicrobial efficacy was observed with the Shozan preparation. The total bacterial contamination of the floor decreased by 80% after treatment with this product, while the contamination on equipment was reduced by 60%. It was established that new environmentally safe disinfectant compositions based on neutral anolyte, shungite, and ozone are highly effective disinfectants.

The results of industrial trials demonstrated that the use of these preparations reliably ensures the decontamination of treated environmental surfaces, even with a minimal exposure time of 30 min. Therefore, these new disinfectants can be used for disinfecting objects under veterinary supervision.

## 4. Discussion

In our studies, the antiseptic formulation “Shozan” demonstrated high bactericidal activity against *B. melitensis*, making it a promising disinfectant for veterinary practice.

Composition and Mechanism of Action:1.The mixture is composed of highly active metastable (electrochemically activated) chlorine–oxygen and hydroperoxide compounds (oxidants). This combination of active substances prevents microorganisms from adapting to biocidal effects. These results align with the findings of Styazhkina et al. (2003), which demonstrated the antiseptic effect of anolyte solution on laboratory rats [17].2.The antibacterial activity of shungite is due to its active components, represented by hydrated fullerenes. They exert anti-inflammatory, bactericidal, and antiseptic effects on both humans and animals. The natural mineral shungite also exhibits bacteriostatic activity against *Streptococcus* and *Staphylococcus*, as confirmed by the studies of Abdulla A.A. [18].3.Ozone (O_3_) is a powerful oxidant that is widely used for the disinfection and decontamination of food products and raw materials [19].

Our studies on the antibacterial activity of ozone, anolyte, propolis, and shungite, as well as their complexes (“Shozan”, “Prozan”, and “Shuprozan”), using the growth inhibition method on saprophytic cultures *E. coli* and *S. aureus* in MPB showed that, in isolated form, these natural substances exhibit low antibacterial activity. However, their complexes demonstrated significantly stronger antibacterial properties, indicating that the combined use of natural antiseptics enhances their effectiveness due to a synergistic effect [9].

Thus, our studies confirmed that the components of “Shozan” exhibit pronounced antibacterial properties, whereas “Prozan” and “Shuprozan” did not show antibacterial activity against pathogenic microorganisms.

At the same time, “Shozan”, “Prozan”, and “Shuprozan” did not demonstrate antiviral activity against the lumpy skin disease virus and foot-and-mouth disease virus, which limits their application in combating viral infections. Furthermore, studies showed the absence of the fungicidal activity of these preparations against *T. verrucosum* and *H. farciminosum*.

It should also be noted that 70% ethanol and 3% hydrogen peroxide did not exhibit antiviral or fungicidal activity.

The comparison with traditional antiseptics, including 70% ethanol and 3% hydrogen peroxide, showed that natural-based preparations are less universal. However, they offer several advantages, including environmental safety and a lower likelihood of microbial resistance development.

“Shozan” also demonstrated high effectiveness in disinfecting surfaces in industrial workshops, reducing the total microbial contamination level by 60–80%.

## 5. Conclusions

Natural antiseptic agents represent a promising direction in veterinary practice due to their high efficacy and environmental safety. This study confirmed that Shozan effectively inhibits the growth of pathogenic bacteria including *B. melitensis* and significantly reduces microbial contamination on industrial surfaces.

Shozan demonstrated several notable advantages, including low toxicity, minimal irritation, and safety for both humans and animals.

Natural antiseptics have the potential to serve as an important alternative to synthetic disinfectants, particularly in light of the increasing resistance of microorganisms to conventional antiseptics. Their application not only ensures effective disinfection, but also contributes to environmental protection. Based on this study’s findings, Shozan, Prozan, and Shuprozan hold significant potential for widespread adoption in veterinary and medical practice, particularly as environmentally friendly disinfection agents.

## Figures and Tables

**Table 1 pathogens-14-00321-t001:** List of microorganism strains used for research.

№	Strain Name
1	*Neethling RIBSP*—the causative agent of lumpy skin disease (LSD) (cattle)—the strain of the lumpy skin disease virus
2	Aphtae epizooticae strain—foot-and-mouth disease virus
3	*B. melitensis* “16M” strain—pathogen (small ruminants)
4	Pasteurella/Saigas/2012/Kostanay/KZ strain—Pasteurellosis pathogen (animals)
5	5G strain (Histoplasma farciminosum)—Lymphangitis pathogen (horses)
6	KO strain (Trichophyton verrucosum)—*Trichophytosis* pathogen (cattle)

**Table 2 pathogens-14-00321-t002:** Microbial growth assessment scale.

Complete Absence of Microbial Growth	Microbial Growth
-	+

**Table 3 pathogens-14-00321-t003:** Fungal growth evaluation scale.

Complete Absence of Fungal Growth	Fungal Growth
-	+

**Table 4 pathogens-14-00321-t004:** Bacteriostatic activity of the preparations against *B. melitensis* and *P. Multocida*.

BSC ^2^, Cells/mL	Shozan ET, min ^3^	Volume of the Shozan Preparation, µL ^1^					
		Shozan	70% EtOH ^4^	H_2_O_2_ ^5^	Shozan	70% EtOH	H_2_O_2_
		*B. melitensis*			*P. multocida*		
**After 24 h**							
1.7 × 10^9^	45	1000- *	1000-	1000-	1000+ *	1000-	1000-
		900-	900-	900-	900+	900-	900-
		800-	800-	800-	800+	800-	800-
		700-	700-	700-	700+	700-	700-
		600-	600-	600-	600+	600-	600-
		500-	500-	500-	500+	500-	500-
		400-	400-	400-	400+	400+	400-
		300-	300-	300-	300+	300+	300-
		200-	200-	200-	200+	200+	200-
		100+	100-	100-	100+	100+	100-
1.7 × 10^9^	30	1000-	1000-	1000-	1000+	1000-	1000-
		900-	900-	900-	900+	900-	900-
		800-	800-	800-	800+	800-	800-
		700-	700-	700-	700+	700-	700-
		600-	600-	600-	600+	600+	600-
		500-	500-	500-	500+	500+	500-
		400-	400-	400+	400+	400+	400-
		300-	300-	300+	300+	300+	300-
		200-	200-	200+	200+	200+	200+
		100+	100-	100+	100+	100+	100+
1.7 × 10^9^	15	1000-	1000-	1000-	1000+	1000+	1000-
		900-	900-	900-	900+	900+	900-
		800-	800-	800-	800+	800+	800-
		700-	700-	700-	700+	700+	700-
		600-	600-	600-	600+	600+	600-
		500-	500-	500-	500+	500+	500-
		400-	400-	400+	400+	400+	400+
		300+	300-	300+	300+	300+	300+
		200+	200-	200+	200+	200+	200+
		100+	100-	100+	100+	100+	100+
Control Bacterial Culture 1.7 × 10^8^ cells/mL		+	+	+	+	+	+
**After 48 h**							
1.7 × 10^9^	45	1000-	1000-	1000-	1000+	1000-	1000-
		900-	900-	900-	900+	900-	900-
		800-	800-	800-	800+	800-	800-
		700-	700-	700-	700+	700+	700-
		600-	600-	600-	600+	600+	600-
		500-	500-	500-	500+	500+	500-
		400-	400+	400-	400+	400+	400-
		300-	300+	300+	300+	300+	300+
		200+	200+	200+	200+	200+	200+
		100+	100+	100+	100+	100+	100+
1.7 × 10^9^	30	1000-	1000-	1000-	1000+	1000-	1000-
		900-	900-	900-	900+	900-	900-
		800-	800-	800-	800+	800+	800-
		700-	700-	700-	700+	700+	700-
		600-	600-	600-	600+	600+	600-
		500-	500-	500+	500+	500+	500-
		400-	400-	400+	400+	400+	400-
		300+	300+	300+	300+	300+	300-
		200+	200+	200+	200+	200+	200+
		100+	100+	100+	100+	100+	100+
1.7 × 10^9^	15	1000-	1000-	1000-	1000+	1000-	1000-
		900-	900-	900-	900+	900+	900-
		800+	800-	800-	800+	800+	800-
		700+	700+	700-	700+	700+	700-
		600+	600+	600+	600+	600+	600-
		500+	500+	500+	500+	500+	500-
		400+	400+	400+	400+	400+	400+
		300+	300+	300+	300+	300+	300+
		200+	200+	200+	200+	200+	200+
		100+	100+	100+	100+	100+	100+
**After 72 h**							
1.7 × 10^9^	45	1000-	1000-	1000-	1000+	1000-	1000-
		900+	900-	900-	900+	900+	900-
		800+	800-	800-	800+	800+	800-
		700+	700-	700-	700+	700+	700-
		600+	600-	600-	600+	600+	600-
		500+	500-	500-	500+	500+	500-
		400+	400+	400-	400+	400+	400-
		300+	300+	300+	300+	300+	300-
		200+	200+	200+	200+	200+	200-
		100+	100+	100+	100+	100+	100-

^1^ Exposure time: minute, ^2^ Bacterial suspension concentration, ^3^ bacterial suspension concentration: cells/mL, ^4^ 70% ethanol, ^5^ hydrogen peroxide; * complete absence of microbial growth, + microbial growth.

**Table 5 pathogens-14-00321-t005:** Antiviral activity of the tested compounds against lumpy skin disease virus.

Compounds	Compound Concentration, µL										
	1000	900	800	700	600	500	400	300	200	100	50
Prozan	++++	++++	++++	++++	++++	++++	++++	++++	++++	++++	++++
Shuprozan	++++	++++	++++	++++	++++	++++	++++	++++	++++	++++	++++
Shozan	++++	++++	++++	++++	++++	++++	++++	++++	++++	++++	++++
70% Ethanol	- *	-	++++	++++	++++	++++	++++	++++	++++	++++	++++
3% H_2_O_2_	++++	++++	++++	++++	++++	++++	++++	++++	++++	++++	++++

* complete absence of viral growth, + weak growth, ++ moderate growth, +++ good growth, ++++ active growth.

**Table 6 pathogens-14-00321-t006:** Antiviral activity of the compounds against the foot-and-mouth disease virus.

Compounds	Compound Concentration, µL										
	1000	900	800	700	600	500	400	300	200	100	50
Prozan	++++ *	++++	++++	++++	++++	++++	++++	++++	++++	++++	++++
Shuprozan	++++	++++	++++	++++	++++	++++	++++	++++	++++	++++	++++
Shozan	++++	++++	++++	++++	++++	++++	++++	++++	++++	++++	++++
70% Ethanol	-	**-**	++++	++++	++++	++++	++++	++++	++++	++++	++++
3% H_2_O_2_	-	**-**	**-**	++++	++++	++++	++++	++++	++++	++++	++++

* complete absence of viral growth, + weak growth, ++++ active growth.

**Table 7 pathogens-14-00321-t007:** Fungicidal activity of the compounds against fungi of the genera *T. verrucosum* and *H. farciminosum*.

Concentration of Fungal Suspension, Cells/mL	Contact Time of the Preparations with the Fungal Suspension, min	Volume of the Compounds µL ^1^					
		Shozan	70% EtOH ^4^	H_2_O_2_ ^5^	Shozan	70% EtOH	H_2_O_2_
		*T. verrucosum*			*H. farciminosum*		
**After 24 h**							
1.7 × 10^9^	45	1000+ *	1000+	1000+	900+	900+	900+
		800+	800+	800+	700+	700+	700+
		600+	600+	600+	500+	500+	500+
		400+	400+	400+	300+	300+	300+
		200+	200+	200+	100+	100+	100+
1.7 × 10^9^	30	1000+	1000+	1000+	900+	900+	900+
		800+	800+	800+	700+	700+	700+
		600+	600+	600+	500+	500+	500+
		400+	400+	400+	300+	300+	300+
		200+	200+	200+	100+	100+	100+
1.7 × 10^9^	15	1000+	1000+	1000+	900+	900+	900+
		800+	800+	800+	700+	700+	700+
		600+	600+	600+	500+	500+	500+
		400+	400+	400+	300+	300+	300+
		200+	200+	200+	100+	100+	100+
**Control fungal culture: 1.7 × 10^9^ cells/mL**		+	+	+	+	+	+

* “+” fungal growth.

**Table 8 pathogens-14-00321-t008:** Antimicrobial efficacy of Shozan, Prozan, and Shuprozan in the production workshops of LLP «Zhaiyk Et».

Test Object	Total Viable Count, CFU per cm^2^	Treatment Result After 30-Minute Exposure, CFU/cm^2^		
		Shozan	Prozan	Shuprozan
Workshop Unit 1—Live Poultry				
Walls	8.1 × 10^4^	1.2 × 10^2^	6.4·× 10^2^	5.4 × 10^2^
Floors	1.9 × 10^6^	3.35 × 10^3^	4.8 × 10^3^	6.8 × 10^3^
Poultry hangers	4.8 × 10^4^	2.12 × 10^4^	4.2 × 10^4^	4.1 × 10^4^
Conveyor (Transporter)	7.39 × 10^3^	6.3 × 10^2^	9.2 × 10^2^	8.2 × 10^2^
Control (untreated)	4.8 × 10^4^	Profuse Growth		
Workshop Unit 2—Stunning and Bleeding				
Walls	9.1 × 10^4^	3.2 × 10^2^	4.5 × 10^3^	6.1 × 10^3^
Floors	2.9 × 10^6^	4.2 × 10^4^	9.2 × 10^5^	8.4 × 10^5^
Poultry hangers	6.8 × 10^4^	6.3 × 10^2^	3.8 × 10^3^	4.3 × 10^3^
Conveyor (Transporter)	9.39 × 10^3^	9.2 × 10^2^	6.4 × 10^3^	7.4 × 10^3^
Control (untreated)	8.1 × 10^4^	Profuse Growth		
Workshop Area 3—Scalding				
Walls	1.2 × 10^4^	4.8 × 10^2^	9.1 × 10^3^	7.8 × 10^3^
Floors	1.5 × 10^4^	8.2 × 10^2^	5.7 × 10^3^	1.8 × 10^3^
Poultry hangers	4.3 × 10^4^	1.8 × 10^2^	9.1 × 10^3^	8.57 × 10^3^
Conveyor (Transporter)	4.3 × 10^4^	1.8 × 10^2^	9.1 × 10^3^	8.57 × 10^3^
Control (untreated)	5.1 × 10^4^	Profuse Growth		
Workshop Area 4—Sanitary Treatment				
Walls	2.8 × 10^6^	7.7 × 10^3^	6.6 × 10^4^	8.2 × 10^3^
Floors	1.9 × 10^6^	9.2 × 10^3^	3.1 × 10^4^	5.4 × 10^3^
Poultry hangers	4.2 × 10^4^	6.4 × 10^2^	2.2 × 10^3^	8.4 × 10^3^
Conveyor (Transporter)	5.1 × 10^4^	3.6 × 10^3^	4.4 × 10^4^	4.6 × 10^4^
Control (untreated)	5.1 × 10^4^	Profuse Growth		
Workshop Area 5—Cutting and Evisceration				
Walls	2.8 × 10^4^	1.7 × 10^3^	1.6 × 10^4^	8.4 × 10^3^
Floors	1.5 × 10^4^	1.2 × 10^3^	1.1 × 10^4^	4.4 × 10^3^
Poultry hangers	3.14 × 10^4^	3.4 × 10^2^	6.2 × 10^3^	9.4 × 10^3^
Conveyor (Transporter)	2.63 × 10^4^	1.6 × 10^3^	1.4 × 10^4^	1.6 × 10^4^
Control (untreated)	2.63 × 10^4^	Profuse Growth		
Workshop Area 6—Packaging				
Walls	2.7 × 10^6^	4.06 × 10^2^	3.14 × 10^4^	3.78 × 10^4^
Floors	1.2 × 10^4^	2.4 × 10^2^	2.63 × 10^3^	8.57 × 10^3^
Packaging Table	2.36 × 10^4^	3.15 × 10^2^	1.54 × 10^3^	7.39 × 10^3^
Control (Untreated)	2.36 × 10^4^	Profuse Growth		

## Data Availability

No new data were generated or analyzed in this study. Data supporting the findings of this study are available from previously published sources, which have been appropriately cited in the manuscript.

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
