# Peer review of "Natural Antiseptics in Veterinary Practice: Evaluation of Efficacy and Safety"

_pathogens, 2025, doi:10.3390/pathogens14040321_

Round 1
Reviewer 1 Report
Comments and Suggestions for Authors
Nametov et al. investigate the antibacterial, antifungal, and antiviral properties of three antiseptic formulations containing natural components, comparing their effectiveness to that of 3% hydrogen peroxide and 70% ethanol.
Below, I present several comments and suggestions regarding the presentation of the study’s results:
- I recommend dividing both the Materials and Methods and Results sections into subsections to organize the text according to specific research objectives. This would improve readability and make the content more accessible to the reader.
- The microbial strains used in the study could be listed and described in a table for clarity.
- Evaluation of antibacterial properties:
- The methodology requires a more detailed description. How were formulations Shozan, Prozan and Shuprozan applied to bacterial cultures? Were they added to liquid cultures or spread on solid media? How was the exposure time to the antiseptic agents terminated? Was post-exposure incubation performed on new plates (i.e., were bacteria transferred after the removal of the antiseptic agent)?
- What criteria were used to assess the experimental outcomes? Does (-) "inhibition of bacterial growth" indicate a complete absence of growth or simply reduced growth compared to the control (without antiseptic exposure)? If it indicates reduced growth, how was this determined—subjectively, visually, or by measuring CFU? If the evaluation was performed subjectively, including representative images of both (+) and (-) results would be valuable.
- The results would be more effectively presented as bar graphs. For instance, bars of uniform height (corresponding to the total antiseptic volume of 1000 µL) could be divided into two segments: one indicating bacterial growth (+) and the other representing inhibition (-).
- If the authors prefer to maintain a tabular format, adjustments to column descriptions are necessary. The table currently presents two sets of columns, and only the title suggests that one corresponds to B. melitensis and the other to P. multocida. Clearer labeling would improve interpretation.
- In the table, under the "after 48 hours" group, the second column from the right (for EtOH) shows a (-) result at 200 but (+) results at 100 and 300. Could this be an error in data placement?
- The discussion does not address the functional role of individual components in the formulations. The compositions of Shozan, Prozan and Shuprozan are very similar, yet only formulation Shozan demonstrated antiseptic properties. Based on the ingredient comparison, one might hypothesize that shungite is primarily responsible for the antibacterial effect, while propolis could have an antagonistic effect, potentially counteracting its activity. While this conclusion may be premature, it would be beneficial to explore how the formulation’s composition influences its effectiveness, particularly since variations in efficacy were observed.
- The reference to Table 1 appears to be misplaced within the text (line 183), where the discussion refers to formulations Prozan and Shuprozan, while the table presents data related to formulation Shozan.
- Evaluation of antifungal properties:
- The methodology requires a more detailed description. How was the exposure time to formulations Shozan, Prozan and Shuprozan terminated? For example, was centrifugation used to remove the antiseptic agent from the cultures?
- What criteria were used to assess the experimental outcome? Did a positive result (indicating inhibition or antifungal activity) mean a complete absence of fungal growth, or merely a reduction compared to the control (without antiseptic exposure)? If the latter, how was this reduction determined—subjectively, visually, or by measuring CFU? If the evaluation was performed subjectively, including representative images of the results would enhance clarity.
- The results of this experiment are missing; only the conclusions have been presented.
- In multiple sections—including the abstract (lines 22–23), results (lines 230–231), and discussion (lines 254–256)—findings and conclusions are presented that do not appear to be the subject of this study. While these statements highlight the advantages of the tested antiseptic agents, if such information is included in the publication, it should be supported by appropriate references to previous studies where these properties were investigated.
Lines 231–234: Missing references.
- When mentioning microbial species for the second time (and thereafter), their names should be written in abbreviated form.
I have no comments regarding the English language. A clear style and appropriate grammar have been used.
Author Response
Thank you for taking the time to review our manuscript. We sincerely appreciate the constructive feedback
provided. Below, we have addressed each comment in detail, and the corresponding revisions or corrections
have been incorporated into the resubmitted files using track changes. We value the reviewers’ insights and
have carefully considered their suggestions to improve the quality of our work
Comments 1: A) I recommend dividing both the Materials and Methods and Results sections into
subsections to organize the text according to specific research objectives. This would improve
readability and make the content more accessible to the reader. B) The microbial strains used in
the study could be listed and described in a table for clarity
Response 1:
A) In the Materials and Methods section, as well as in the Research Results section, subsections
have been introduced based on microorganism groups (bacteria, viruses, fungi).
B) The microbial strains used in the study are presented in Table 1. Line -119.
Comments 2:
Evaluation of antibacterial properties:
2.1 The methodology requires a more detailed description. How were formulations Shozan,
Prozan and Shuprozan applied to bacterial cultures? Were they added to liquid cultures or spread
on solid media? How was the exposure time to the antiseptic agents terminated? Was postexposure incubation performed on new plates (i.e., were bacteria transferred after the removal of
the antiseptic agent)?
2.2 What criteria were used to assess the experimental outcomes? Does (-) "inhibition of
bacterial growth" indicate a complete absence of growth or simply reduced growth compared to
the control (without antiseptic exposure)? If it indicates reduced growth, how was this
determined—subjectively, visually, or by measuring CFU? If the evaluation was performed
subjectively, including representative images of both (+) and (-) results would be valuable.
2.3 The results would be more effectively presented as bar graphs.
For instance, bars of uniform height (corresponding to the total antiseptic volume of 1000 µL)
could be divided into two segments: one indicating bacterial growth (+) and the other
representing inhibition (-). If the authors prefer to maintain a tabular format, adjustments to
column descriptions are necessary.
2.4 The table currently presents two sets of columns, and only the title suggests that one
corresponds to B. melitensis and the other to P. multocida. Clearer labeling would improve
interpretation.
2.5 The compositions of Shozan, Prozan and Shuprozan are very similar, yet only formulation
Shozan demonstrated antiseptic properties. Based on the ingredient comparison, one might
hypothesize that shungite is primarily responsible for the antibacterial effect, while propolis could
have an antagonistic effect, potentially counteracting its activity. While this conclusion may be
premature, it would be beneficial to explore how the formulation’s composition influences its
effectiveness, particularly since variations in efficacy were observed.
2.6 The reference to Table 1 appears to be misplaced within the text (line 183), where the
discussion refers to formulations Prozan and Shuprozan, while the table presents data related to
formulation Shozan.
2.7 In the table, under the "after 48 hours" group, the second column from the right (for EtOH)
shows a (-) result at 200 but (+) results at 100 and 300. Could this be an error in data placement?
The discussion does not address the functional role of individual components in the formulations.
3
Response 2:
2.1 Methodology: We have expanded the research methods for evaluating the effects of antiseptic agents. Liquid medicinal forms in the form of infusions were used, and the preparations were added to bacterial culture suspensions. To determine their effects, solid nutrient media (MPA—meat-peptone agar) were used.The impact of antiseptic agents occurs throughout the entire laboratory study. The post-exposure
incubation period involves transferring the samples onto meat-peptone agar.
2.2 Were bacterial growth criteria used?
"-" indicates complete absence of microbial growth, and "+" indicates any microbial growth.
Microbial growth was determined visually by multiple researchers.Line -149 table 2.
2.3 We found that using a table is more effective, as it includes multiple parameters (bacterial species, exposure time, antiseptic types and doses, and bacterial suspension concentration). We did not present the data as bar graphs because there is a large amount of data, which would make the graph overly complex and difficult to interpret. Instead, we kept the data in a detailed table format, as shown in Table 7 (Fungicidal activity of the compounds against fungi of the genus T. verrucosum and H. farciminosum), to ensure clarity and easy comparison Line-339.
2.4 We modified the column descriptions to include bacterial species (Brucella melitensis,
Pasteurella multocida). We corrected a technical error in Table 4.
2.5 In the Discussion section, we described the mechanism of action of the components of the antiseptic agents. Lines -381-392.
2.6 We separated the description of the antiseptic agents Prozan, Ozan, and Shuprozan from the description of the Shozan agent, which is linked in Table 1. Lines -113-117.
2.7 We apologize for the oversight in the table where we failed to mention the pathogens. In the revised version of the table, the pathogens have been clearly added. The columns are now more clearly labeled to indicate that one set corresponds to B. melitensis and the other to P. multocida, ensuring improved clarity for better interpretation. Line 284 (table 4).
Comment 3:
Evaluation of antifungal properties:
•The methodology requires a more detailed description. How was the exposure time to
formulations Shozan, Prozan and Shuprozan terminated? For example, was centrifugation
used to remove the antiseptic agent from the cultures?
• What criteria were used to assess the experimental outcome? Did a positive result
(indicating inhibition or antifungal activity) mean a complete absence of fungal growth, or
merely a reduction compared to the control (without antiseptic exposure)? If the latter,
how was this reduction determined—subjectively, visually, or by measuring CFU? If the
evaluation was performed subjectively, including representative images of the results
would enhance clarity.
• The results of this experiment are missing; only the conclusions have been presented.
Response 3:
3.1 Methodological Enhancements
Additional details were incorporated into the Methodology section regarding the study of fungi. The inoculation on agar was performed without centrifugation of the fungal culture and antiseptic mixture. Line -210.
3.2 Evaluation of Fungicidal Properties
To assess the fungicidal properties of antiseptics, a fungal growth assessment scale was used, as outlined in Table 3. Line -214.
The results were evaluated visually by multiple researchers. The presence or absence of fungal colony growth was determined solely by visual assessment.
3.3 Presentation of Fungicidal Activity Results
The findings on fungicidal activity are summarized in Table 7. Line 339.
Comment 4 additional comments:
• In multiple sections—including the abstract (lines 22–23), results (lines 230–231), and
discussion (lines 254–256)—findings and conclusions are presented that do not appear to be the
subject of this study. While these statements highlight the advantages of the tested antiseptic
agents, if such information is included in the publication, it should be supported by appropriate
references to previous studies where these properties were investigated.
• Lines 231–234: Missing references.
• When mentioning microbial species for the second time (for example, B. melitensis
instead Brucella melitensis ), their names should be written in abbreviated form.
Response 4:
4.1 Text Revisions
After reviewing the text, we agreed that certain statements were not directly related to the results of our study. These statements have been removed from the following sections:
- Lines 22–23 (general text)
- Lines 230–231 (results)
- Lines 254–256 (discussion)
4.2 Correction of Lines 231–234
Lines 231–234 have been revised and corrected:
- Inaccurate statements were removed
- References were deemed unnecessary and deleted.
4.3 Adjustments to Microorganism Mentions
The feedback was accepted, and mentions of microorganism species were reduced throughout the text.

Reviewer 2 Report
Comments and Suggestions for Authors
In this study, the authors evaluated the efficacy and safety of natural antiseptic formulations in combating some pathogenic microorganisms and their potential application as environmentally friendly disinfectants.
It is a paper well done but has some shortcomings such as the fact that a very limited number of pathogens were tested.
I ask you to follow some of my suggestions.
The abstract cites the names of the formulations used without alluding to their composition. Delete the names and write what they are made of (es. shungite, propolis, ozone and anolyte).
Line 74: Add “some” before “pathogenic microorganisms”
Line 86: Add "mainly" before "in small ruminants".
Line 104: Add reference
Lines 108 and 114: Add “antiseptic” before “formulation”
Line 118: Only 45 min? Previously you also talked about 15 and 30 minutes.
Line 145: How many ml does a spray dispersion correspond to? How large is the surface area for a spray dispersion? This data is important.
The results and table are presented unclearly. In Table 1, what data are related to Brucella melitensis and what data are related to Pasteurella multocida?
Line 151 : Add “The results obtained using Shozan are shown in Table 1”.
Line 183: Delete “table 1”
The results of the microbial composition in the production LLP «Zhaiyk Et» are presented in a very superficial way. Dedicate a separate subchapter and go into more detail.
Line 231: Add reference.
Line 236-237: Remove the sentence.
Comments on the Quality of English Language
I cannot evaluate the quality of English Language.
Author Response
Thank you for taking the time to review our manuscript. We sincerely appreciate the constructive feedback provided. Below, we have addressed each comment in detail, and the corresponding revisions or corrections have been incorporated into the resubmitted files using track changes. We value the reviewers’ insights and have carefully considered their suggestions to improve the quality of our work
Comment 1: The abstract cites the names of the formulations used without alluding to their
composition. Delete the names and write what they are made of (es. shungite, propolis, ozone
and anolyte).
Response 1: The composition of the antiseptic agents used has been added to their names in
the abstract. Lines: 17, 20,21.
Comment 2:
Line 74: Add “some” before “pathogenic microorganisms”
Response 2: The phrase was corrected by adding "some" before “pathogenic microorganisms. Line 73.
Comment 3:
Line 86: Add "mainly" before "in small ruminants".
Response 3: The text was adjusted by adding "mainly" before "in small ruminants. Line114.
Comment 4:
Line 104: Add reference
Response 4: A reference was added as required. Line -160.
Comment 5:
Lines 108 and 114: Add “antiseptic” before “formulation”
Response 5: The term "antiseptic" was added before “formulation” in both instances. Line -161.
Comment 6:
Line 118: Only 45 min? Previously you also talked about 15 and 30 minutes.
Response 6: The methodology for studying the antiviral activity of the preparations involves an
exposure time of only 45 minutes.
Comment 7:
Line 145: How many ml does a spray dispersion correspond to? How large is the surface area
for a spray dispersion? This data is important.
Response 7: The Materials and Methods section has been supplemented with a methodology for
disinfecting rooms and equipment using antiseptic agents. The application was carried out by
mechanical spraying at a spray flow rate of 1 liter per 5 m². Line -223.
Comment 8:
The results and table are presented unclearly. In Table 1, what data are related to Brucella
melitensis and what data are related to Pasteurella multocida?
Response 8 We have revised the column descriptions to include the microbial species Brucella
melitensis and Pasteurella multocida. Line – 284, table -4.
Comment 9:
Line 151: Add “The results obtained using Shozan are shown in Table 1”.
3
Response 9: The correction has been made in Table 4 (previously referenced as Table 1) Line -
281.
Comment 10:
Line 183: Delete “table 1”
Response 9: Agreed with the comment, the correction has been made. Line -277. (The text was
revised because the paragraph originally mentioned Shozan, Shuprozan, and other
antiseptics, while Table 1 included data only on Shozan. We removed references to other
antiseptics, keeping only Shozan, and after the revision, Table - 1 was renumbered as Table
4.)
Comment 11:
The results of the microbial composition in the production LLP «Zhaiyk Et» are presented in
a very superficial way. Dedicate a separate subchapter and go into more detail.
Response 11: The microbial composition results from the "Zhaiyk Et" production facility have
been described in a separate subsection. Line 350.
Comment 12:
Line 231: Add reference.
Response 12: We have removed the paragraph.
Comment 13:
Line 236-237: Remove the sentence.
Response 13: We have removed the paragraph containing information comparing antiseptics
with other antiseptics. The statement in the paragraph referred to resistance, but it was not
supported by experiments or references. Therefore, we removed the paragraph.

Reviewer 3 Report
Comments and Suggestions for Authors
1 Line 33:Considering the use of “increase” instead of “rise” in this sentence for clarity and precision.
2 Line52:Changing “negatively impact cardiovascular function” to “negative impact on cardiovascular function” for improved clarity and consistency in phrasing.
3 Line77-79: The antiseptic formulations Shuprozan (shungite + propolis + ozone + anolyte), Prozan (propolis + ozone + anolyte), and Shozan (shungite + ozone + anolyte) are listed only by name without specifying the exact amounts or proportions of each component. To enhance the reproducibility and scientific rigor of the study, it is recommended that the authors provide the specific amounts or proportions of each ingredient, or at least a range, to help readers better understand the effectiveness and mechanisms of these formulations.
4 Line224:Using “the contamination” instead of “that of ”makes it clearer and more specific.
5 Line231-232: Change “identified in relation to hydrogen peroxide”with “observed with hydrogen peroxide”.
6 Line236: Changing “The study results confirm” to “The study results confirmed” for tense consistency.
7 Line250: Uses “such as” instead of “including”for smoother flow.
8 The discussion section mentions the significant efficacy of natural antiseptic agents such as Shozan, Prozan, and Shuprozan but lacks references to other studies for comparison. To strengthen the argument, I recommend including relevant literature that supports or contrasts with your findings. Additionally, a comparison with other antimicrobial agents would further enhance the discussion.
9 Line 262: changing “including” to “such as” would improve the flow of the sentence.
10 Line 267-269: The study results show that Shozan has significant potential, but Prozan and Shuprozan did not demonstrate notable antimicrobial effects. Therefore, the conclusion stating that all three agents hold significant potential for widespread adoption may need to be adjusted to better reflect the study findings.
Comments on the Quality of English Language
The English expression requires slight modification
Author Response
Thank you for taking the time to review our manuscript. We sincerely appreciate the constructive feedback provided. Below, we have addressed each comment in detail, and the corresponding revisions or corrections have been incorporated into the resubmitted files using track changes. We value the reviewers’ insights and have carefully considered their suggestions to improve the quality of our work.
Comments 1: 1 Line 33:Considering the use of “increase” instead of “rise” in this sentence for clarity
and precision.
Respond 1:
We have replaced the word "rise" with "increase". Line -31.
Comments 2: Line52:Changing “negatively impact cardiovascular function” to “negative impact on
cardiovascular function” for improved clarity and consistency in phrasing.
Respond 2: We have replaced the word. Line -50.
Comments 3: Line77-79: The antiseptic formulations Shuprozan (shungite + propolis + ozone + anolyte),
Prozan (propolis + ozone + anolyte), and Shozan (shungite + ozone + anolyte) are listed only by name
without specifying the exact amounts or proportions of each component. To enhance the reproducibility and scientific rigor of the study, it is recommended that the authors provide the specific amounts or proportions of each ingredient, or at least a range, to help readers better understand the effectiveness and mechanisms of these formulations.
Respond 3: The specific quantitative composition of the antiseptic agents and the methods of
their preparation are provided in the Materials and Methods section. Line -80-112.
Comments 4: 4 Line224:Using “the contamination” instead of “that of ”makes it clearer and more
specific.
Respond 4: We have replaced the word. Line -370.
Comments 5: 5 Line231-232: Change “identified in relation to hydrogen peroxide”with “observed with
hydrogen peroxide”.
Respond 5: We have replaced the word. The statement in the paragraph referred to resistance,
but it was not supported by experiments or references. Therefore, we removed the paragraph.
Comments 6 Line236: Changing “The study results confirm” to “The study results confirmed” for tense
consistency.
Respond 6: We paraphrased the paragraph, and as a result, these changes were no longer necessary. Line
376-377.
Comments 7: Line250: Uses “such as” instead of “including”for smoother flow.
Respond 7: We have replaced the word. Line-409.
Comments 8 The discussion section mentions the significant efficacy of natural antiseptic agents such as Shozan, Prozan, and Shuprozan but lacks references to other studies for comparison. To strengthen the argument, I recommend including relevant literature that supports or contrasts with your findings.
Additionally, a comparison with other antimicrobial agents would further enhance the discussion.
Respond 8: References to other studies on the presented antiseptic agents are provided in other sections of the article. Comparisons with traditional antiseptic agents are also included in other sections.
Comments 9: Line 262: changing “including” to “such as” would improve the flow of the sentence.
Respond 9: We have replaced the word. Line – 418.
3
Comments 10: Line 267-269: The study results show that Shozan has significant potential, but Prozan
and Shuprozan did not demonstrate notable antimicrobial effects. Therefore, the conclusion stating that all three agents hold significant potential for widespread adoption may need to be adjusted to better reflect the study findings.
Respond 10: We have made corrections and adjustments in the Conclusion section to more accurately
reflect the results. Line -385, 390, 392.

Round 2
Reviewer 2 Report
Comments and Suggestions for Authors
The authors improved the paper, responding exhaustively to my requests.
Thanks